# Patterns of Extrapulmonary Metastases in Sarcoma Surveillance

**DOI:** 10.3390/cancers13184669

**Published:** 2021-09-17

**Authors:** Zachery Hong, Patrick England, Lee Rhea, Angela Hirbe, Douglas McDonald, Cara A. Cipriano

**Affiliations:** 1Washington University School of Medicine, St. Louis, MO 63110, USA; zacheryhong@wustl.edu (Z.H.); pengland@wustl.edu (P.E.); 2United States Environmental Protection Agency, Washington, DC 20460, USA; rhea.lee@epa.gov; 3Division of Oncology, Department of Medicine, Washington University School of Medicine, St. Louis, MO 63110, USA; hirbea@wustl.edu; 4Division of Musculoskeletal Oncology, Department of Orthopaedic Surgery, Washington University School of Medicine, St. Louis, MO 63110, USA; mcdonaldd@wustl.edu; 5Division of Orthopaedic Oncology, Department of Orthopaedic Surgery, Perelman School of Medicine, University of Pennsylvania, Philadelphia, PA 19104, USA

**Keywords:** sarcoma, surveillance, extrapulmonary, metastases

## Abstract

**Simple Summary:**

Soft tissue sarcomas are cancerous tumors that can cause death by spreading throughout the body (metastasizing), most commonly to the lungs. After initial removal of the tumor, patients undergo periodic chest imaging for 10 years to monitor for metastases, but experts disagree about whether the abdomen/pelvis should be imaged as well. Our goal was to determine when, where, and how frequently metastases occur in the abdomen/pelvis; this would help answer the question of whether these areas should be imaged as part of surveillance. We found that 7% of patients treated for soft tissue sarcoma developed abdomen/pelvis metastases. Compared to lung metastases, abdomen/pelvis metastases appeared later and over a broader period of time. We were not able to predict which patients are most likely to develop abdomen/pelvis metastases. Understanding these patterns will help experts develop surveillance guidelines that will identify metastatic cancer without subjecting patients to more scans than necessary.

**Abstract:**

Soft tissue sarcomas (STS) most commonly metastasize to the lungs. Current surveillance guidelines variably recommend abdominal and pelvic imaging, but there is little evidence to support this. We sought to determine the proportion of initial pulmonary versus extrapulmonary metastases, the time to development of each, and factors to identify patients that would benefit from abdominopelvic surveillance. We retrospectively reviewed 382 patients who underwent surgical treatment for STS at a single institution. Of the 33% (126/382) of patients who developed metastases, 72% (90/126) were pulmonary, 22% (28/126) were extrapulmonary, and 6% (8/126) developed both simultaneously. Initial extrapulmonary metastases occurred later (log rank *p* = 0.049), with median 11 months (IQR, 5 to 19) until pulmonary disease and 22 months (IQR, 6 to 45) until extrapulmonary disease. Pulmonary metastases were more common in patients with high grade tumors (*p* = 0.0201) and larger tumors (*p* < 0.0001). Our multivariate analysis did not identify any factors associated with initial extrapulmonary metastases. A substantial minority of initial metastases were extrapulmonary; these occurred later and over a broader time range than initial pulmonary metastases. Moreover, extrapulmonary metastases are more difficult to predict than pulmonary metastases, adding to the challenge of creating targeted surveillance protocols.

## 1. Introduction

Soft tissue sarcomas (STS) are a group of rare, histologically diffuse neoplasms of mesenchymal origin. STS can originate anywhere in the musculoskeletal or peripheral nervous system, and local treatment typically includes wide resection and radiation. Even after appropriate management of the primary tumor, STS have the potential to metastasize, leading to a mortality rate of 60% in the first five years after treatment [1]. Because of this, post-treatment surveillance is universally practiced, and prognostic factors such as histologic grade, location, size, and time since treatment are used to dictate the type and frequency of imaging based on the likelihood of metastases [2,3].

The lungs are the initial site of metastases in 80% of patients who develop distant disease [4,5,6], with extrapulmonary metastases occurring less frequently, and often after pulmonary metastases have been identified [4,7]. Although pulmonary surveillance is well-accepted, the use of computed tomography (CT) of the abdomen and pelvis for routine sarcoma surveillance remains controversial, with little evidence to support this practice [5,6,8,9]. In addition to the relatively low incidence of extrapulmonary metastases, there are also concerns related to unnecessary diagnostic imaging, including radiation exposure, incidental findings requiring additional workup, and increased health care expenditures [10]. Finally, while certain histologic subtypes of STS are linked with increased risk of extrapulmonary metastases, no other independent prognostic factors have been identified thus far [6,7]. As such, current guidelines for patients with STS are more variable and less specific regarding surveillance of the abdomen and pelvis [7,11,12,13,14].

Given the uncertain indications for routine extrapulmonary surveillance in patients with STS, we sought to answer the following questions: (1) Of the patients who developed metastatic disease following initial treatment, what proportion presented with extrapulmonary metastases alone compared to pulmonary metastases or both? (2) Was time from surgery to initial metastasis different for pulmonary versus extrapulmonary metastases? (3) Can patient characteristics (age, body mass index [BMI] and gender) and pathologic factors (histologic type, grade, depth, location and size) be used to identify patients at increased risk for developing pulmonary versus extrapulmonary metastases?

## 2. Materials and Methods

We performed a retrospective review including all patients who underwent resection of a non-metastatic STS by one of two orthopedic oncologists at a single sarcoma center between August 1999 and December 2018. Approval was obtained from the institutional review board at Washington University School of Medicine, and informed consent was obtained from all patients. A total of 545 patients with STS received treatment at our institution during the study period. This included 36 patients with positive margins from a prior excision who were referred to our institution for definitive surgical treatment. Thirty-five patients with low grade liposarcomas and 21 patients with metastatic STS at presentation were excluded. Additionally, 107 patients were lost to follow up within 2 years of initial treatment and were not known to have died, leaving 382 patients available for study. (Figure 1) Patients that were lost to follow up were compared with included patients on the basis of relevant patient and tumor characteristics. The two groups differed in terms of age, as younger patients were more likely to follow up, and surgical margin status, as patients with positive margins were more likely to follow up compared to patients with widely negative (≥5 mm) margins. (Table 1)

Patients were managed according to standard sarcoma treatment and surveillance guidelines. All patients underwent resection of their primary STS. Adjuvant radiation therapy was used for high grade tumors and myxoid sarcomas, with the exception of those that were small and superficial with wide margins, as well as for patients that underwent amputation with wide margins. Radiation was used for patients with low grade sarcomas when widely negative margins could not be achieved surgically. Overall, 87% (333/382) of patients underwent radiation therapy, including 94% (303/323) of patients with high grade STS and 51% (30/59) of patients with low grade STS. Patients were subsequently followed by a multidisciplinary sarcoma team consisting of orthopedic, medical and radiation oncologists. High grade sarcomas were followed every 6 months for 3 to 5 years; while low grade sarcomas were followed every 6 months until 5 years, then annually thereafter. Routine surveillance included physical examination, local magnetic resonance imaging (MRI), and CT of the chest, abdomen and pelvis, with occasional use of positron emission tomography (PET) in conjunction with CT if indicated. Potential metastatic findings on surveillance imaging received shorter-interval follow up and/or biopsy as indicated.

### 2.1. Variables and Outcome Measures

Pulmonary metastases were confined to the lung parenchyma and pleura, while metastases occurring in any other visualized tissue were considered extrapulmonary. Diagnoses of metastases were confirmed with imaging or biopsy. Independent variables related to the primary tumor included depth, size, grade, histopathology, location and surgical margins. Tumor depth was characterized relative to investing fascia. Tumor sizes were based on their greatest dimension and categorized into three groups: ≤5 cm, 5 to 10 cm, and >10 cm. All STS were graded as low, intermediate, or high based on the French Federation of Cancer Centers Sarcoma Group (FNCLCC) system, and tumors were grouped according to histologic type (Table 2). Finally, surgical margins were categorized into four groups that represented: positive margins, malignancy at <0.1 cm from the nearest margin, malignancy at <0.5 cm from the nearest margin, and malignancy at ≥0.5 cm from the nearest margin.

### 2.2. Demographics and Description of Distant Metastases

Thirty-three percent (126/382) of patients developed distant metastases. The distribution of location for initial metastases was: 72% (90 patients) pulmonary, 22% (28 patients) extrapulmonary, and 6% (8 patients) with simultaneous pulmonary and extrapulmonary metastases. The median age of patients at the time of diagnosis with extrapulmonary metastases was 59 years (IQR, 44 to 73) versus 60 years (IQR, 40 to 71) for pulmonary metastases. Eighteen patients with extrapulmonary metastases died of their disease, seven within 24 months of initial treatment. The median length of follow-up for patients with extrapulmonary metastases was 38 months (IQR, 22 to 79). (Table 3).

Twelve out of 15 histologic subtypes had development of initial extrapulmonary metastases. Eighty-nine percent (25/28) of tumors were high grade and 82% (23/28) were deep to the investing fascia. The most common locations for initial extrapulmonary metastases were the lymph nodes and appendicular bone, followed by the liver. (Table 4) Out of the 36 initial extrapulmonary metastases that were detected, 19 were found with abdominopelvic CT and 12 were found with routine chest CT. The remaining 5 extrapulmonary metastases were in the head and distal extremities, and were detected using extremity MRI, whole body PET or head CT.

### 2.3. Statistical Analysis

To answer our first study question, the proportion of patients who developed initial extrapulmonary metastases was determined. A binomial proportion test was performed to compare the proportion of patients with metastatic STS who developed initial extrapulmonary versus initial pulmonary metastases.

To answer our second study question, whether there was a difference in time to extrapulmonary versus pulmonary metastases following primary tumor resection, a Kaplan–Meier curve was created, and a log rank test was used to compare metastases-free survival between these groups. Median time to initial metastases for each group was also calculated and compared using a Kruskal–Wallis test.

To answer our third study question, whether the development of initial extrapulmonary metastases was associated with clinically relevant factors (patient age, BMI, and gender, tumor grade, depth, location, histologic type and size) or a combination of these variables, a t-test or Kruskal–Wallis test (as appropriate) was performed for continuous variables, and a chi-square test (or Fischer’s exact test for comparisons with inadequately populated cases) was performed for categorical variables.

To address transfer bias in our study, included patients were compared to those who were lost to follow up in terms of tumor (size, location, grade, depth, surgical margins) and patient (age, gender, BMI) characteristics. This analysis was performed using t-tests or Shapiro–Wilk tests for continuous variables, and chi-square or Fisher exact tests for categorical variables. Median age was greater in patients who were lost to follow up than those who were included in the study 63 (IQR, 50 to 76) and 58 (IQR, 42 to 70), respectively; *p* = 0.002)), and, patients with widely negative margins were more likely to be lost to follow up than patients with positive margins [Odds Ratio (OR) = 1.96, 95% confidence interval (CI): 1.11–3.46, *p* < 0.01] (Table 1). 

SPSS statistical software (IBM Corp, Armonk, NY, USA) was used to perform all statistical analyses. A significance threshold of 0.05 was considered to be statistically significant.

## 3. Results

### 3.1. Incidence of Initial Extrapulmonary Metastases

Thirty-three percent (126/382) of patients developed distant metastases. Of these, 72% (90/126) had initial metastatic disease located in the lungs, while 22% (28/126) had extrapulmonary metastases. The remaining 6% (8/126) of patients were found to have simultaneous pulmonary and extrapulmonary metastases. These results show that initial extrapulmonary metastases occur less commonly than initial pulmonary metastases (binomial proportion 0.22 [95% CI 0.15 to 0.30]; *p* < 0.001).

### 3.2. Time to Development of Initial Extrapulmonary Metastases

Initial extrapulmonary metastases occurred later than initial pulmonary metastases (log rank *p* = 0.049) (Figure 2). Median time from surgical treatment to development of metastases was 11 months (IQR, 5 to 19) for isolated pulmonary and 22 months (IQR, 6 to 45) for isolated extrapulmonary disease (*p* = 0.08) (Table 3).

### 3.3. Association of Relevant Patient and Tumor Characteristics with Extrapulmonary Metastases

Our multivariate regression was unable to detect independent prognostic factors that would be useful in identifying patients at an increased risk of developing new extrapulmonary metastases. Patients with high grade tumors were at a greater risk of developing pulmonary metastases compared to those with low grade tumors [OR = 2.916, 95% CI: 1.183–7.191, *p* = 0.0201]. In addition, increasing tumor size was found to correlate with an increased incidence of new pulmonary metastases (OR = 4.414, 95% CI: 2.245–8.680, *p* < 0.0001 for tumor size <5 cm versus 5–10 cm; OR = 3.530, 95% CI: 1.778–7.008, *p* = 0.0003 for tumor size <5 cm versus >10 cm) (Table 5).

## 4. Discussion

Surveillance following initial treatment is considered an essential component of sarcoma management. The goal is to detect recurrent or metastatic disease in order to provide patients with prognostic information and treatment. Sarcomas most commonly metastasize to the lungs, so the importance of pulmonary surveillance is universally accepted; however, extrapulmonary surveillance remains controversial. This knowledge gap has led to differences in surveillance recommendations and practices between geographic regions and among individual providers. We sought to build on the existing literature by studying a large cohort of STS patients to better define the incidence and timing of extrapulmonary metastases, as well as any factors that could identify patients more likely to develop extrapulmonary metastases and therefore benefit from abdominopelvic imaging.

### 4.1. Limitations

There are several limitations to this work. First, while our study found that routine abdominopelvic imaging may lead to earlier diagnosis of metastases, it was not designed to assess potential morbidity and mortality benefits of extrapulmonary imaging. While metastatic STS was historically considered untreatable, strategies for managing systemic disease have emerged over recent decades with novel immunotherapy modalities being particularly exciting [15]. Research on pulmonary metastasectomy has shown an association between fewer pulmonary nodules, greater resectability, extended disease-free interval, and improved survival [8,16]. Studies evaluating locally ablative modalities, such as stereotactic body radiotherapy and cryoablation, have also produced promising results in improving progression-free survival and local control of pulmonary metastases. These modalities may play a future role in treatment for extrapulmonary metastases due to their quicker recovery times and greater accessibility to lesions in the bone, liver, and visceral sites compared to surgical intervention [17,18]. However, we are not aware of any data demonstrating clear morbidity and mortality benefits to early treatment of extrapulmonary disease. Other relevant issues, such as the time and cost associated with closer surveillance, are also beyond the scope of this study—as such, we cannot draw definitive conclusions or recommendations about whether abdominopelvic imaging should be included in routine surveillance.

Second, because sarcoma is an uncommon disease and extrapulmonary metastases are even more rare, our sample size was limited. This may have caused our analysis to be underpowered with respect to detecting associations between the clinicopathological variables in our multivariate model and the development of new extrapulmonary metastases.

Our study was also limited by its retrospective design in several respects. Each follow-up visit included physical examination, chest CT, and local MRI, in accordance with current guidelines. The lack of clear recommendations for extrapulmonary surveillance may have led to more variable acquisition of abdominal and pelvic CT. This may have resulted in a sampling bias, with some patients developing extrapulmonary metastases that were not imaged before their pulmonary metastases were diagnosed; in this case, our results would underestimate the frequency of initial extrapulmonary disease. Transfer bias limited our study as well, as 22% of patients received less than 2 years of follow-up and were excluded from the analysis. Older patients were more likely to be lost to follow up, although the difference of median 63 compared to 58 years is not likely to be clinically significant in this context. Patients with widely negative (>0.5 cm) surgical margins were also more likely to be lost to follow up compared to those with positive margins. A potential explanation for this could be that patients with positive surgical margins were aware of their higher risk for recurrence, causing them to be more consistent with follow up. As we did not identify any risk factors for extrapulmonary metastases, we have no indication of how this may have changed our findings. Finally, the evolution of sarcoma treatment over the course of our 20-year study period may have led to co-treatment bias; however, given the complexity of these factors and their intersectionality, we are unable to determine their effect on our results.

### 4.2. Existing Literature

#### 4.2.1. Proportion of Initial Extrapulmonary Metastases 

Our study found that 33% of patients developed distant metastases after resection of their primary STS. Within this group, 22% were initially diagnosed with extrapulmonary metastases, while an additional 6% of patients were found to have simultaneous pulmonary and extrapulmonary metastases. The overall frequency of distant metastases in our patient population is comparable with that reported in the literature, which ranges between 20% and 40% [2,4,5,7,9,19,20,21,22]. The rarity of this condition has limited prior studies, thus contributing to the lack of evidence-based guidelines. The majority of existing studies report 20 or fewer incidences of isolated extrapulmonary metastases, averaging approximately 5% of all STS patients [2,4,5,6,7,22,23,24]. Because of these low numbers, authors have generally refrained from providing recommendations regarding extrapulmonary surveillance. Some have concluded that routine imaging of the abdomen and pelvis is difficult to support given the low incidence of initial extrapulmonary metastases and the interplay of other variables such as cost of imaging, as well as complications associated with false positives and related work-up [7,24]; others have recommended that all patients be given the option to receive CT of the abdomen and pelvis as part of their regular follow-up.

Studies on STS patients followed with CT of the chest, abdomen, and pelvis have found that extrapulmonary metastases are rare relative to pulmonary metastases, and they tend to occur after pulmonary metastases have been detected. One study reported that 10% (14/140) of patients had abdominal or pelvic metastases; however, only four of these patients first developed isolated extrapulmonary metastases, while the other 10 patients developed extrapulmonary metastases after pulmonary metastases had already been diagnosed [7]. The other study reported that 16% (20/124) of patients developed extrapulmonary metastases, but only seven patients had initial extrapulmonary metastases. These studies found the incidence of initial extrapulmonary metastases to be 3% and 6%, respectively, compared to the 7% incidence we observed. An additional eight patients in our study were initially diagnosed with pulmonary and extrapulmonary metastases, making it unclear which developed first. Also of note, 12 of the 36 initial extrapulmonary metastases were detected on chest CT, so it would have been identified without abdominopelvic imaging.

Although low compared to the frequency of pulmonary metastases in STS patients, a 7% incidence has warranted surveillance in other malignancies. For example, asynchronous lung metastases occur in 1.7% to 7.2% of patients with colorectal cancer [25], and surveillance guidelines for this population generally include routine CT of the chest every 6 to 12 months for the first 3 to 5 years after surgery [26,27]. In patients with stage IV colorectal cancer, resection of pulmonary metastases is not curative but has been shown to significantly improve 5-year survival (13.8% in all patients with pulmonary metastases, but 32–61% in patients able to undergo pulmonary metastectomy) [25]. A similar improvement in survival has not been clearly defined for resection of extrapulmonary STS metastases, although new advances in treatment have shown promising preliminary results [15,17,18].

Overall, our findings suggest that abdominal and pelvic imaging in routine surveillance may lead to earlier diagnosis of metastases in a substantial minority of patients. Of note, the incidence of initial extrapulmonary metastases in STS is comparable to that of metastases routinely screened for in other cancers; however, a clear survival benefit of early detection of extrapulmonary metastases in STS has not yet been demonstrated.

#### 4.2.2. Time to Development of Initial Extrapulmonary Metastases

Our study found a longer interval of metastases-free survival between patients who developed initial extrapulmonary compared to pulmonary metastases. The median time from surgical treatment to diagnosis of initial extrapulmonary metastases was twice that of initial pulmonary metastases (22 and 11 months, respectively); moreover, the IQR was over twice as broad for median time to extrapulmonary metastases (6 to 45) compared to pulmonary metastases (5 to 19).

Prior literature on the timing of metastatic diagnosis in STS has focused on pulmonary disease. Most of these metastases are identified within the first two years of STS diagnosis, especially in patients with high grade tumors [14,22,24,28]; hence, the NCCN and ESMO surveillance guidelines recommend more frequent imaging in the first 2–3 years for patients with high and intermediate grade STS. [12,13]. However, these intervals may not accurately reflect the timing for extrapulmonary metastases. Thompson et al. reported an average time to isolated extrapulmonary metastases of 12 months among four patients; this is only half the time we observed, but the discrepancy may be related to the small number in their study and the large range of time to extrapulmonary metastases in ours. Among the 28 patients with isolated extrapulmonary metastases in our study, the median time to diagnosis was 22 months after surgery; moreover, 25% of these initial metastases were not diagnosed until ≥45 months. Thus, while extrapulmonary metastases were less common, they were also less predictable in terms of timing, which limits the development of targeted surveillance strategies.

#### 4.2.3. Factors Associated with Extrapulmonary versus Pulmonary Metastases

Previous studies have found increased depth, higher grade, larger size, and shorter length of the disease-free interval to be associated with distant metastases in general [5,9,29], but not specifically to extrapulmonary metastases. To our knowledge, there are no prior studies which have investigated prognostic factors for the development of extrapulmonary metastases in STS. While our data confirmed that patients with larger, higher grade tumors experienced an increased incidence of new pulmonary metastases, we were unable to identify patient or tumor variables that could be used to identify patients with an increased risk of developing extrapulmonary metastases. Specifically, our multivariate analysis did not detect an association between patient age, gender, BMI, tumor size, depth, grade, location, or histologic subtype with development of initial extrapulmonary metastases. Thus, although higher grade and larger size have been widely linked to increased risk of metastases in STS [6,19,20,21,29], we did not observe either of these variables to be associated with initial extrapulmonary metastases.

Prior studies have demonstrated a relatively high incidence of extrapulmonary metastases in patients with myxoid liposarcomas. Estourgie et al. and Muratori et al. found that 55% of metastases in patients with myxoid liposarcomas were extrapulmonary, and both groups of authors emphasized the importance of routine extrapulmonary imaging for surveillance [30,31]. Current NCCN guidelines name angiosarcomas, epithelioid sarcomas, leiomyosarcomas, and myxoid liposarcomas as histologic subtypes with increased frequency of extrapulmonary metastases [13] but do not cite evidence supporting this recommendation. While we did not detect increased risk associated with histologic subtype in our multivariate analysis, 67% (2/3) of angiosarcoma patients, 17% (1/6) of epithelioid sarcoma patients, and 17% (3/18) of myxoid liposarcomas patients developed initial extrapulmonary metastases; by contrast, only 3% (2/66) of patients with metastatic leiomyosarcomas had initial extrapulmonary metastases. While histologic diagnosis did not reach significance in our regression model, the high incidence of initial extrapulmonary metastases among patients with certain sarcoma subtypes warrants further investigation in a larger patient population. 

## 5. Conclusions

Of the 33% of patients with primary STS that developed metastatic disease after resection, 22% were initially diagnosed with extrapulmonary metastases, and an additional 6% with simultaneous pulmonary and extrapulmonary metastases. Initial extrapulmonary metastases occurred later and over a broader time range than pulmonary metastases. While patients with larger, higher grade tumors were more likely to develop pulmonary metastases, we were unable to identify patient or tumor variables that were associated with an increased risk of extrapulmonary metastases. These findings suggest that routine abdominal and pelvic imaging may detect new metastatic disease in a substantial minority of patients. Future studies with larger numbers may identify risk factors for extrapulmonary disease, but at this time we are unable to predict which, if any, patients are more or less likely to benefit from routine abdominopelvic imaging. Further research is needed to evaluate the potential advantages of early detection of extrapulmonary metastases on treatment and survival. Additionally, better understanding the psychosocial impact of increased imaging, the potential morbidity of investigating false positive findings, and the overall cost-effectiveness of advanced imaging will be necessary to develop definitive surveillance recommendations.

## Figures and Tables

**Figure 1 cancers-13-04669-f001:**
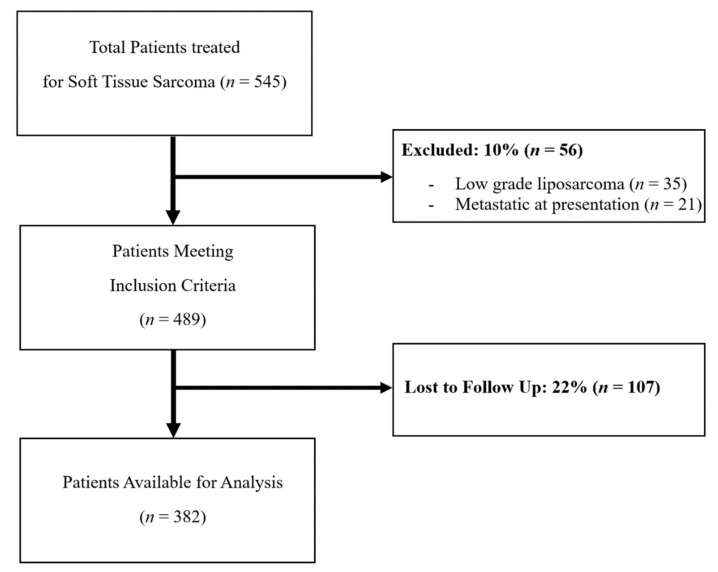
This STROBE flow chart demonstrates the inclusion of patients in this study.

**Figure 2 cancers-13-04669-f002:**
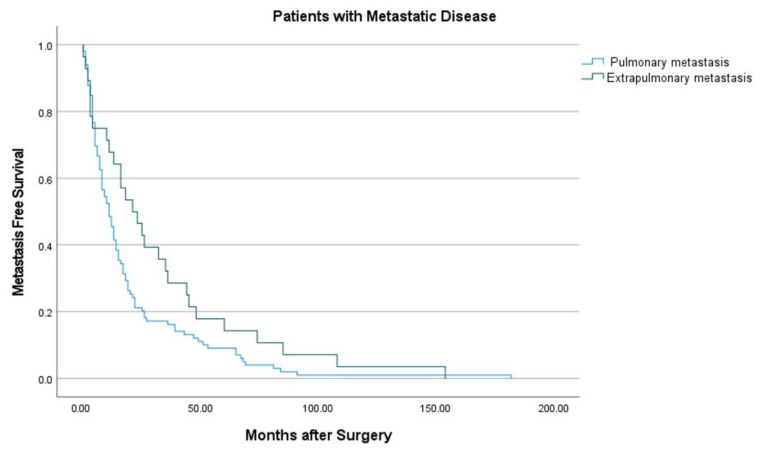
Kaplan–Meier curves for metastasis-free survival demonstrate that initial extrapulmonary metastases occurred later than initial pulmonary metastases (log rank *p* = 0.049).

**Table 1 cancers-13-04669-t001:** Distribution of lost to follow up and included patients’ clinical and tumor characteristics.

Parameter	Included Patients (*n* = 382)	Patients Lost to Follow Up (*n* = 107)	Odds Ratio (95% CI)	*p*-Value
Patient Gender				0.66 ^e^
Female	44 (167)	41 (44)	-	
Male	56 (215)	59 (63)	1.11 (0.72–1.72)	
Patient Age (years) ^b^	58.0 (42.0, 70.0)	63.0 (50.0, 76.0)	-	0.002 ^d^
Patient BMI (kg/m^2^) ^a^	28.9 ± 6.7	28.9 ± 6.9	-	0.98 ^c^
Largest Tumor Dimension (cm)	7.0 (4.0, 11.3)	7.5 (4.5, 15.0)	-	0.34 ^d^
Categorical Tumor Size				0.65 ^d^
≤5 cm	37 (140)	41 (44)	-	
5–10 cm	32 (123)	30 (32)	0.89 (0.53–1.50)	
>10 cm	31 (119)	29 (31)	0.89 (0.53–1.51)	
Tumor Location				0.22 ^e^
Lower Extremity	58 (222)	67 (72)	-	
Upper Extremity	17 (66)	18 (19)	0.89 (0.50–1.58)	
Back/Torso/Neck	10 (37)	3 (4)	0.33 (0.11–0.97)	
Pelvis/Sacrum	9 (33)	6 (6)	0.56 (0.23–1.39)	
Acral (hand/foot)	6 (24)	6 (6)	0.77 (0.30–1.96)	
Tumor Depth				0.74 ^e^
Superficial	13 (49)	11 (12)	-	
Deep	87 (333)	89 (94)	1.15 (0.57–2.48)	
Tumor Grade				0.60 ^d^
Low	15 (59)	17 (18)	-	
Intermediate	2 (7)	3 (3)	1.43 (0.33–6.10)	
High	83 (316)	80 (86)	0.91 (0.51–1.62)	
Tumor Margins				0.01 ^d^
Positive Margins	28 (106)	22 (23)	-	
<0.1 cm	23 (89)	18 (19)	0.98 (0.50–1.92)	
<0.5 cm	20 (76)	18 (19)	1.09 (0.55–2.16)	
≥0.5 cm	29 (108)	42 (45)	1.96 (1.11–3.46)	

Data are presented as ^a^ mean ± standard deviation for parametric data, or ^b^ median (IQR) for nonparametric data or column % (*N*). *p*-values: c = ANOVA, d = Kruskal–Wallis test, e = Fisher’s Exact test.

**Table 2 cancers-13-04669-t002:** Patient, tumor, and treatment characteristics.

Parameter	All Patients % (*n*) Total = 382
Patient Gender	
Female	44 (167)
Male	56 (215)
Patient Age (years) ^b^	58.0 (42.0–70.0)
Patient BMI (kg/m^2^) ^a^	28.9 ± 6.7
Largest Tumor Dimension (cm)	7.0 (4.0–11.3)
Categorical Tumor Size	
≤5 cm	37 (140)
5–10 cm	32 (122)
>10 cm	31 (120)
Tumor Location	
Lower Extremity	58 (222)
Upper Extremity	17 (66)
Back/Torso/Neck	10 (37)
Pelvis/Sacrum	9 (33)
Acral	6 (24)
Tumor Depth	
Superficial	13 (49)
Deep	87 (333)
Tumor Grade	
Low	15 (59)
Intermediate	2 (7)
High	83 (216)
Tumor Margins	
Positive Margins	28 (106)
<0.1 cm	23 (89)
<0.5 cm	20 (76)
≥0.5 cm	29 (108)
Tumor Histopathology	
Undifferentiated Pleomorphic Sarcoma	30.4 (116)
Leiomyosarcoma	17.3 (66)
Fibrosarcoma	11.5 (44)
Sarcoma, Unspecified	9.7 (37)
Synovial Sarcoma	8.1 (31)
Liposarcoma	6.5 (25)
Myxoid Liposarcoma	4.7 (18)
Malignant Peripheral Nerve Sheath Tumor	3.1 (12)
Epithelioid Sarcoma	1.6 (6)
Rhabdomyosarcoma	1.6 (6)
Chondrosarcoma/Osteosarcoma	1.3 (5)
Ewing/Round Cell Sarcoma	1.3 (5)
Alveolar/Embryonal Rhabdomyosarcoma	1.0 (4)
Alveolar Soft Part Sarcoma	0.8 (3)
Angiosarcoma	0.8 (3)
Hemangiopericytoma	0.3 (1)
Time to metastasis (months) ^b^	-
Length of follow-up (months) ^b^	44.9 (26.8–75.5)

Data are presented as ^a^ mean ± standard deviation for parametric data, or ^b^ median (IQR) for nonparametric data or column % (*n*).

**Table 3 cancers-13-04669-t003:** Distribution of the metastatic soft tissue sarcoma patients’ clinical and tumor characteristics.

Parameter	Extrapulmonary Metastases (*n* = 28)	Pulmonary Metastases (*n* = 90)	Odds Ratio (95% CI)	*p*-Value
Patient Age (years) ^b^	57.5 (39.3–67.8)	59.0 (45.0–69.0)	-	0.41 ^d^
Patient BMI (kg/m^2^) ^a^	29.0 ± 7.0	30.4 ± 6.8	-	0.35 ^c^
Largest Tumor Dimension (cm) ^b^	7.1 (4.6–10.2)	8.5 (6.0–12.5)	-	0.12 ^d^
Categorical Tumor Size				0.26 ^d^
≤5 cm	29 (8)	16 (14)	-	
5–10 cm	46 (13)	46 (41)	0.59 (0.21–1.72)	
>10 cm	25 (7)	39 (35)	0.38 (0.12–1.22)	
Tumor Location				0.69 ^e^
Lower extremity	54 (15)	60 (54)	-	
Upper extremity	18 (5)	14 (13)	1.41 (0.43–4.58)	
Back/torso/neck	7 (2)	11 (10)	0.73 (0.14–3.71)	
Pelvis/sacrum	18 (5)	9 (8)	2.29 (0.65 to 8.04)	
Acral (hand/foot)	4 (1)	6 (5)	0.73 (0.08 to 6.76)	
Tumor Depth				0.13 ^e^
Superficial	18 (5)	6 (5)	-	
Deep	82 (23)	94 (85)	3.04 (0.67 to 13.21)	
Tumor Grade				0.67 ^d^
Low	11 (3)	7 (6)	-	
Intermediate	-	1 (1)	-	
High	89 (25)	92 (83)	0.70 (0.17–2.92)	
Tumor Margins				0.26 ^d^
Positive	29 (8)	20 (18)	-	
<0.1 cm	36 (10)	27 (24)	0.94 (0.31 to 2.85)	
<0.5 cm	7 (2)	22 (20)	0.22 (0.04 to 1.20)	
≥0.5 cm	29 (8)	31 (28)	0.62 (0.20 to 1.95)	
Months to metastasis ^b^	22 (6–45)	11 (5–19)	-	0.08 ^d^
Months of follow up ^b^	38 (22–79)	31 (16–49)	-	0.50 ^d^
Months from first operation until death ^b^	30 (12–48)	24 (13–39)	-	0.31 ^d^

Data are presented as ^a^ mean ± standard deviation for parametric data, ^b^ median (IQR) for nonparametric data, or column % (*n*). *p*-values: c = ANOVA, d = Kruskal–Wallis test, e = Fisher’s Exact test.

**Table 4 cancers-13-04669-t004:** Locations of extrapulmonary metastases.

Location	% (*n*)
Lymph Node	25 (9)
Bone, Appendicular	25 (9)
Liver	14 (5)
Bone, Axial	8 (3)
Chest Wall	5 (2)
Adrenal Gland	5 (2)
Spleen	3 (1)
Pancreas	3 (1)
Small Bowel	3 (1)
Brain	3 (1)
Muscle	3 (1)
Femoral Vein	3 (1)

**Table 5 cancers-13-04669-t005:** Odds ratio estimates with 95% confidence intervals and *p*-value from the multivariate model on pulmonary metastases.

Parameter (Reference)	Odds Ratio (95% CI)	*p*
Tumor Grade		
High (Low)	2.92 (1.18–7.19)	0.02
Categorical Size		
5–10 cm (<5 cm)	4.41 (2.25–8.68)	<0.001
>10 cm (<5 cm)	3.35 (1.78–7.01)	<0.001
BMI (kg/m^2^)	1.04 (1.00–1.08)	0.03

## Data Availability

The data presented in this study are available on request from the corresponding author. The data are not publicly available due to inclusion of protected health information.

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
