# Peer review of "Patterns of Extrapulmonary Metastases in Sarcoma Surveillance"

_cancers, 2021, doi:10.3390/cancers13184669_

Round 1

Reviewer 1 Report

Well-researched and well-written article

Author Response

Point 1: "Well-researched and well-written article"

Response 1: Thank you for for taking the time to review our paper.

Reviewer 2 Report

Hong et al. retrospectively reviewed 382 patients with STS and aimed to identify tumor or patient characteristics that may predict the initial occurence of extrapulmonary metastases.

The study is concisely written and clearly structured.

Due to the low incidence of extrapulmonary metastases (n = 28), the authors were unable to identify factors associated with initial extrapulmonary metastases. The findings obtained (e.g. extrapulmonary metastases occured later and over a wider period of time compared to pulmonary metastases) are mostly in line with published data.

The study seems underpowered and therefore fails to reach its aims. Moreover, inconclusive recommendations might have contributed to a sampling bias as patients were not properly screened for extrapulmonary metastases until pulmonary metastases occured, as discussed by the authors.

The limitation of this study is mainly the retrospective design leading to substantial sampling and tansfer biases. However, this is far off the scope of any fair revision process. Moreover, the limitations are extensively discussed by the authors.

Apart from this obvious conceptual weakness, I cannot detect any technical mistakes. The conclusions are certainly supported by the results and are not overstated.

It is to be decided by the Editor, if this study represents a valuable resource for future meta analyses.

Author Response

Point 1: "Due to the low incidence of extrapulmonary metastases (n = 28), the authors were unable to identify factors associated with initial extrapulmonary metastases. The findings obtained (e.g. extrapulmonary metastases occured later and over a wider period of time compared to pulmonary metastases) are mostly in line with published data.

The study seems underpowered and therefore fails to reach its aims. Moreover, inconclusive recommendations might have contributed to a sampling bias as patients were not properly screened for extrapulmonary metastases until pulmonary metastases occured, as discussed by the authors."

Response 1: We agree with the reviewer's assessment of the limitations/biases affecting our study and addressed these in our discussion section. The majority of previous studies on extrapulmonary metastases in soft-tissue sarcoma report even fewer incidents of isolated extrapulmonary metastases. This lack of data has been a major barrier in developing evidence-based guidelines for extrapulmonary metastases screening. We hope to add our data to the current literature to increase the power of future meta-analyses. We also characterize some patterns of extrapulmonary metastases, such as time to development and location, which are not well described in the current literature.

Point 2: "The limitation of this study is mainly the retrospective design leading to substantial sampling and tansfer biases. However, this is far off the scope of any fair revision process. Moreover, the limitations are extensively discussed by the authors."

Response 2: We agree with the reviewer's assessment. We address how our study's retrospective design weakens our findings. However, given the rarity of the outcome under investigation, a retrospective case-control design was the only feasible option.

Reviewer 3 Report

Concratulation to authors. This paper is excellent. 

Author Response

Point 1: "Concratulation to authors. This paper is excellent. "

Response 1: Thank you for reviewing our paper.